# Environment-sensitive emission of anionic hydrogen-bonded urea-derivative–acetate-ion complexes and their aggregation-induced emission enhancement

Masaki Takahashi [1✉], Nozomu Ito[1], Naoki Haruta [2,3,4], Hayato Ninagawa[1], Kohei Yazaki[1], Yoshihisa Sei[5], Tohru Sato [2,3,4] & Makoto Obata[1]

Anions often quench fluorescence (FL). However, strong ionic hydrogen bonding between fluorescent dyes and anion molecules has the potential to control the electronic state of FL dyes, creating new functions via non-covalent interactions. Here, we propose an approach, utilising ionic hydrogen bonding between urea groups and anions, to control the electronic states of fluorophores and develop an aggregation-induced emission enhancement (AIEE) system. The AIEE ionic hydrogen-bonded complex (IHBC) formed between 1,8-diphenylnaphthalene (*p*-2Urea), with aryl urea groups at the para-positions on the peri-phenyl rings, and acetate ions exhibits high environmental sensitivities in solution phases, and the FL quantum yield (QY) in ion-pair assemblies of the IHBC and tetrabutylammonium cations is more than five times higher than that of the IHBC in solution. Our versatile and simple approach for the design of AIEE dye facilitates the future development of environment-sensitive probes and solid-state emitting materials.

[1] Interdisciplinary Graduate School of Medicine and Engineering, University of Yamanashi, 4-4-37 Takeda, Kofu, Japan. [2] Fukui Institute for Fundamental Chemistry, Kyoto University, Takano-Nishihiraki-cho 34-4, Sakyo-ku, Kyoto 606-8103, Japan. [3] Department of Molecular Engineering, Graduate School of Engineering, Kyoto University, Nishikyo-ku, Kyoto 615-8510, Japan. [4] Unit of Elements Strategy Initiative for Catalysts & Batteries, Kyoto University, Nishikyo-ku, Kyoto 615-8510, Japan. [5] Laboratory for Chemistry and Life Science, Institute of Innovative Research, Tokyo Institute of Technology, 4259 Nagatsuta, Midori-ku, Yokohama 226-8503, Japan. ✉email: tmasaki@yamanashi.ac.jp

light-emitting organic compounds have been known to emit only in dilute solution, with quenching occurring in the solid-state due to intermolecular interactions[1]. However, in recent years, organic compounds that emit fluorescence (FL) in solid-state form but are quenched in dilute solution, called aggregation-induced emission (AIE) dyes, and those for which the emission quantum yield (QY) increases upon aggregation, aggregation-induced emission enhancement (AIEE) dyes, have been discovered[2,3]. Recently, these compounds have been intensively studied by photochemists and materials scientists from the perspective of fundamental scientific interest and for their potential technological importance in applications such as light-emitting diodes[4], fluorogenic probes for explosive detection[5], monitoring environmental hazards[6], viscosity sensing[7], and bioimaging probes[8]. Typical strategies for the development of efficient solid-state light-emitting molecules include the introduction of a bulky substituents to prevent intermolecular interactions[9], the formation of J-aggregates[10,11], the introduction of electron-donating or electron-withdrawing groups[4,12–14], and the introduction of the molecular structures that exhibit excited-state intramolecular proton transfer[15,16].

However, existing light-emitting efficiencies are not high, and there is a lack of diversity in the currently available molecular skeletons. Indeed, tetraphenyl ethane, the most typical AIE dye, having a simple structure, has a low FL QY of 15% in the aggregate states[17]. Therefore, the development of next-generation solid-state light-emitting organic materials by utilising non-covalent interactions has been attempted in recent years. Only a few successful demonstrations of solid-state emission induced by non-covalent interactions have been reported. These include a coordination bond between the p-orbital of borane compounds and the oxygen atom of aldehydes[18], acid–base interactions[19], halogen–bonding interactions[20], hydrogen–bonding interactions[21], isolation of fluorescent dyes by ionic lattices[22] and an anion–$\pi^+$ interaction[23]. The introduction of electron-donating and electron-withdrawing groups to π-conjugated molecular systems with rotatable covalent bonds is a strategy for AIEE dye synthesis[4,12,13,24]. Molecules having the donor–π–acceptor (D–π–A) structure often show viscosity-sensitive fluorescence and AIEE because of the suppression of a weakly emitting twisted intramolecular charge transfer (TICT) state, resulting in emission from a locally excited (LE) state or planar intramolecular charge transfer (PICT) state, in the solid state or in highly viscous solvents[25]. A contrasting strategy for controlling the electronic state of functional groups is to utilise non-covalent interactions. The addition of a Brønsted or Lewis acid to fluorescent dye with a Lewis basic functional group results in an increase in the electron-withdrawing character of the functional group[26,27] and the addition of a molecular anion to a fluorescent dye with a urea or pyrrole group produces a strongly ionic hydrogen-bonded complex (IHBC), resulting in an increase in the electron-donating character of the functional group[28]. Even when the electronic state of the molecule is altered by these non-covalent interactions, the compounds can still potentially exhibit AIEE characteristics. In this study, we focused on the strong ionic hydrogen-bonding interaction between urea groups and anions and attempted to create non-covalent AIEE materials (Fig. 1).

Recently, fluorescent urea compounds have been studied to examine their interaction with anionic molecules. Because the FL colour changes of the urea compounds occur upon adding anions, these studies have mainly been conducted in solution-phase experiments. This behaviour has been characterised for the development of fluorescent molecular sensors for the detection of specific anionic molecules[28–31]. Moreover, the interaction between anions and fluorescent urea compounds in solution has been considered to be the cause of fluorescence quenching[32,33], and the light-emission characteristics of fluorescent urea derivatives in the solid-state have almost never been examined[34]. We have developed a molecular design for the two urea substituents at the ortho-positions on the peri-phenyl rings on the naphthalene ring (p-2Urea) based on the characteristics of the molecular structures of conventional AIEE dyes. Such dyes typically contain π-electron systems that connect two electron-donating groups, each of which are linked to the π-system via a rotatable single bond[8], or they include π-systems with sterically hindered phenyl groups (Fig. 1)[2]. In addition, the 1-phenyl naphthalene derivative p-1Urea, having a less sterically hindered phenyl group, and m-2Urea, a positional isomer of p-2Urea in which the urea groups are less conjugated with naphthyl group, were also synthesised to investigate the emission characteristics affected by ionic hydrogen bonding between the urea groups and molecular anion and by ion-pair complexation with tetrabutylammonium cations. By examining the properties of these compounds, we found that some IHBCs show AIEE and environmental responsiveness in solution phases.

## Results and discussion

p-1Urea was synthesised using a previously reported method[34]. The synthesis and characterisation of p-2Urea and m-2Urea are described in the Supplementary Method 1 and 2, respectively. In addition, single crystals of p-2Urea were obtained by vapour diffusion of diethyl ether into a DMSO solution. Single-crystal X-ray diffraction analysis demonstrated that p-2Urea forms a hydrogen-bonded solvate complex with DMSO. Oxygen atoms in DMSO interact with two NH groups in urea, forming bifurcated hydrogen bonds (Supplementary Fig. 5). The CIFs file is in Supplementary Data 1.

Combined density functional theory and time-dependent density functional theory (DFT/TD-DFT) calculations for the three compounds and the IHBC of the urea groups and anions were carried out. For the TD-DFT calculations, the geometries of the three urea derivatives without acetate anions were first optimised at the B3LYP/6–31G (d, p) level of theory. Then, using the optimised geometries, excitation energies were calculated at the CAM-B3LYP/6–31G + (d) level of theory[35,36]. All calculations were performed by using the Gaussian 16 package[37] and a common practice with DFT calculations is to replace tert-butyl groups with methyl groups to simplify the calculations and decrease the computational time. The geometries of the IHBC were optimised before excitation energy calculations were carried out at the CAM-B3LYP/6–31G + (d) level of theory. The computational details are described in the Supplementary Method 3 and the atomic coordinates of the optimised molecular geometries are shown in Supplementary Data 2. The results of these calculations are shown in Supplementary Fig. 8. p-2Urea and the IHBC of p-2Urea and acetate are expected to possess different degrees of conjugation and different electronic structures. The computed highest occupied molecular orbital (HOMO) and lowest unoccupied molecular orbital (LUMO) of p-2Urea without AcO− were found to be delocalised over the entire π-system. In contrast, the computed HOMO of the p-2Urea–acetate IHBC was localised on the phenyl urea group substituent of the naphthyl system, and the LUMO of the complex was localised on the naphthyl rings. In addition, the HOMO energy level of the IHBC is increased by the ionic hydrogen bonding between the urea groups and acetate anions. A red shift of the UV absorption spectrum was predicted by TD-DFT calculations due to the charge-transfer character of the anionic p-2Urea–acetate complex (Supplementary Fig. 9 and Supplementary Table 2). Therefore, control of the electronic states of fluorescent molecules by non-covalent bonding and the realisation of AIEE characteristics and viscosity-sensitive fluorescence by switching between PICT and TICT fluorescence are expected for the IHBC.

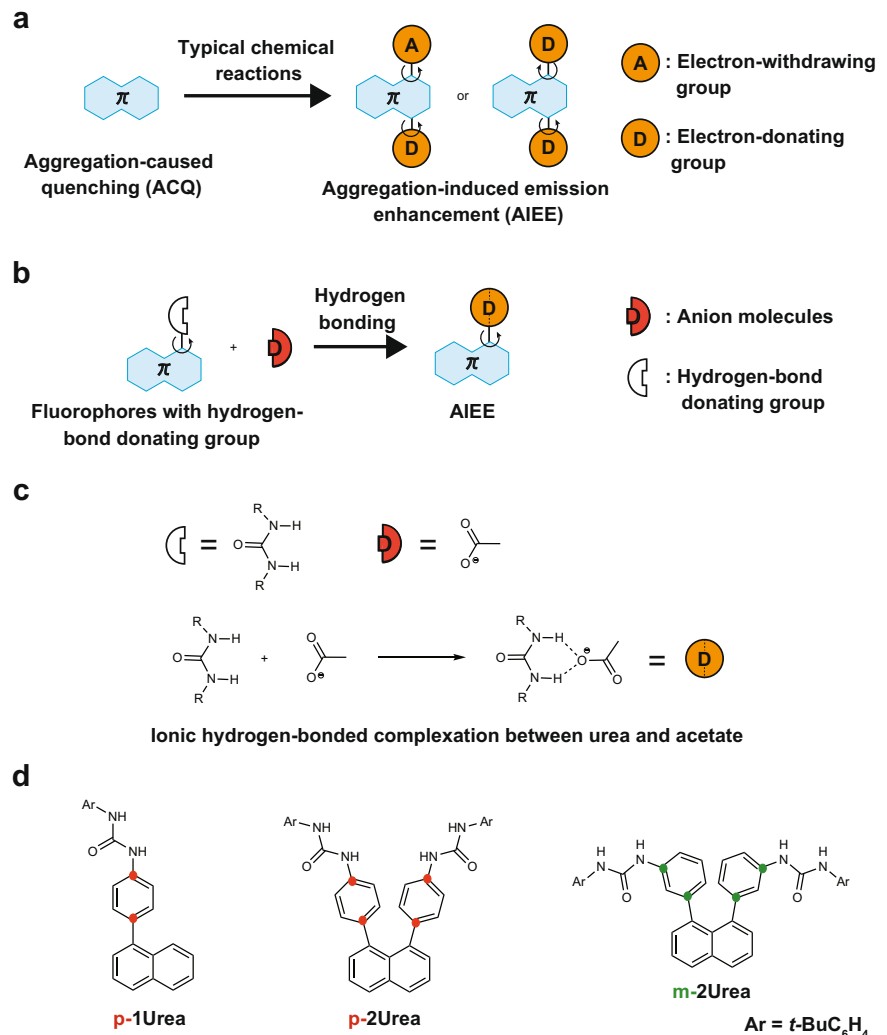

**Fig. 1 Schematic representation of design strategy for ionic hydrogen-bonded complexes (IHBC) with aggregation-induced emission enhancement (AIEE). a** Schematic illustration of the addition of electron-donating (D) and/or electron-withdrawing (A) groups to fluorophores to synthesise typical AIEE fluorophores. **b** Design strategy for IHBC with viscosity-sensitive fluorescence and AIEE (this work). **c** Urea as hydrogen-bond donating group and acetate as hydrogen-bond acceptor. **d** Chemical structures of model molecules: ***p*-1Urea**, ***p*-2Urea**, and ***m*-2Urea**.

**Investigation of hydrogen bonding interactions by ¹H NMR and IR spectroscopies.** In order to examine the ionic hydrogen bonding between the synthesised compound and acetate anions in solution and in the solid-state, the evolution of the solution-phase ¹H NMR spectrum upon the addition of tetra-butylammonium acetate (TBAAc) was observed, and an IR spectrum of the solid-state complex was acquired.

First, the shifting of the ***p*-2Urea** N–H proton peaks in the ¹H NMR spectrum was confirmed upon adding TBAAc to ***p*-2Urea** in DMSO-$d_6$. The N–H proton peaks were shifted from approximately 8.3 ppm to approximately 11.5 ppm and became saturated (Supplementary Fig. 10). From these observations, it was deduced that the electron densities of the N–H protons in the urea group were reduced by the ionic hydrogen bonding with acetate in DMSO. It was previously established that urea compounds form strong ionic hydrogen bonds with acetate, even in highly polar solvents such as DMSO that are thought to inhibit weak hydrogen bonding[38].

In addition, FT-IR spectra were measured to confirm that the acetate and urea groups formed ionic hydrogen bonds, even in the solid-state. As a result, the stretching vibration of the carbonyl of the urea group was shifted from 1651 to 1699 cm⁻¹ upon the

addition of 2 equiv of TBAAc (Supplementary Fig. 11). This result demonstrates that the urea groups of ***p*-2Urea** form intermolecular hydrogen bonds, and the C=O bond in the urea group is weakened in the solid-state without TBAAc. In addition, ionic hydrogen bonding between the N–H protons of the urea groups and acetate ions occurs with the addition of TBAAc, and the weaker intermolecular hydrogen bonds between different urea groups are eliminated with the C=O bond in the urea groups becoming stronger[39]. From these results, it can be concluded that ***p*-2Urea** forms an ionic hydrogen bond with acetate ions, both in DMSO and in the solid state.

**Investigation of changes in the UV–visible absorption and fluorescence spectra upon the addition of TBAAc in solution.** Changes in the ultraviolet–visible (UV–Vis) absorption spectrum of ***p*-2Urea** in DMSO (150 μM) upon addition of TBAAc were investigated. The UV–Vis spectrum of ***p*-2Urea** in DMSO shows maxima at 270 and 340 nm. Upon addition of up to 250 equiv of TBAAc, a red shift in the absorption of ***p*-2Urea** and isosbestic points at 280 and 310 nm were observed (Fig. 2a). The red shift may be attributed to the smaller HOMO-LUMO gap due to the ionic hydrogen bonding between ***p*-2Urea** and acetate ions. Upon

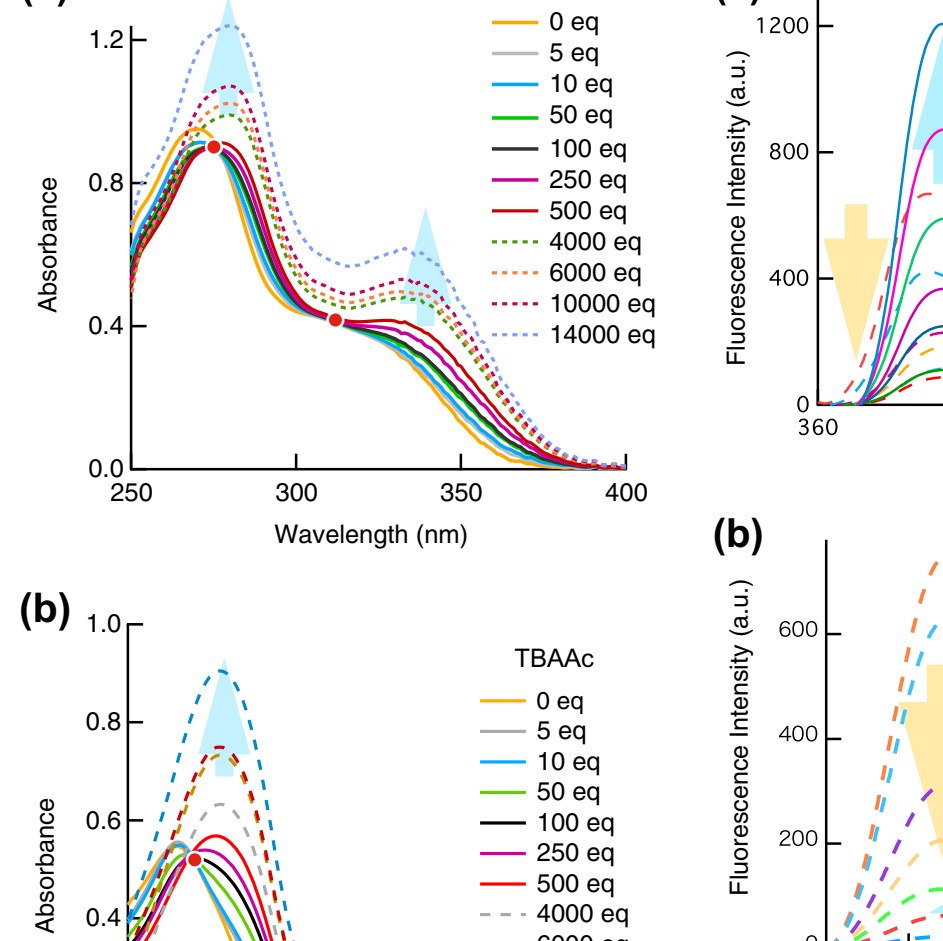

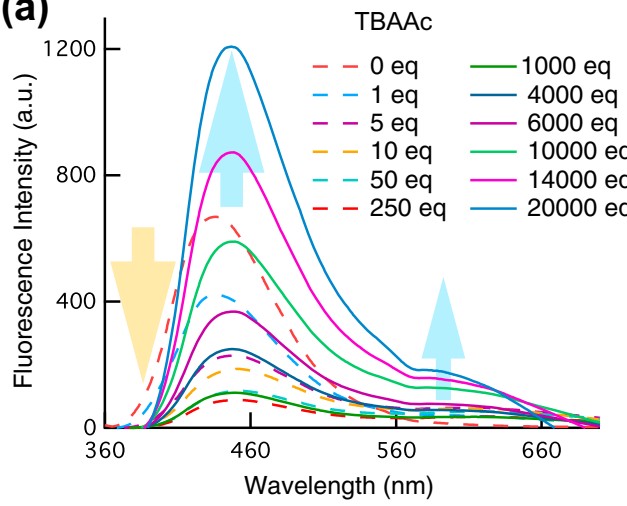

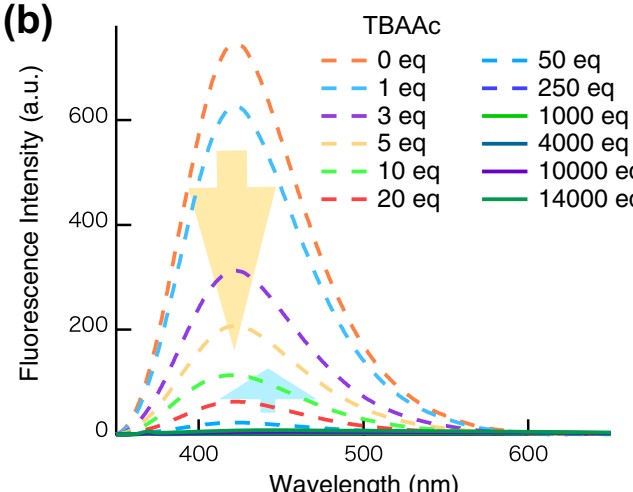

**Fig. 3 Changes in fluorescence spectra. a** Fluorescence spectra of a DMSO solution of ***p*-2Urea** (150 μM, $\lambda_{ex}$ = 359 nm) in the absence and presence of different amounts of tetrabutylammonium acetate (TBAAc) (orange dashed = 0 eq; light blue dashed = 1 eq; purple dashed = 5 eq; yellow dashed = 10 eq; yellowish green dashed = 50 eq; red dashed = 250 eq; green = 1000 eq; dark blue = 4000 eq; purple = 6000 eq; yellowish green =10,000 eq; pink = 14,000 eq; blue = 20,000). **b** Fluorescence spectra of a DMSO solution of ***m*-2Urea** (150 μM, $\lambda_{ex}$ = 332 nm) in the absence and presence of different amounts of TBAAc (orange dashed = 0 eq; light blue dashed = 1 eq; purple dashed = 3 eq; yellow dashed = 5 eq; yellowish green dashed = 10 eq; red dashed = 20 eq; blue dashed = 50 eq; blue purple dashed = 250 eq; yellowish green = 1000 eq; dark blue = 4000 eq; purple = 10,000 eq; green = 14,000).

**Fig. 2 Changes in UV-visible absorption spectra.** Absorbance spectra of **a *p*-2Urea** and **b *m*-2Urea** in the absence and presence of different amounts of tetrabutylammonium acetate (TBAAc) (orange = 0 eq; grey = 5 eq; light blue = 10 eq; yellowish green = 50 eq; green = 100 eq; purple = 250 eq; red = 500 eq; grey dashed = 4000 eq; brown dashed = 6000 eq; reddish brown dashed = 10,000 eq; light blue dashed = 14000 eq).

further addition of excess TBAAc, an increase in the absorbance of ***p*-2Urea** over a wide wavelength range was observed, and the isosbestic points were lost. Further studies are necessary to disclose the causes of the increased absorbance upon the addition of excess amounts of TBAAc.

In contrast, the UV–Vis spectrum of ***m*-2Urea** in DMSO includes a single maxima, at 264 nm. Upon addition of TBAAc (up to 250 equiv), the red shift of the longer absorption region that was seen for ***p*-2Urea** was not observed, and only one isosbestic point, at 270 nm, was apparent (Fig. 2b).

The molecular-orbital calculation results support the hypothesis that the difference in the change in absorption wavelength between the positional isomers depends on the ease of charge transfer along the long-axis direction of the molecule, from the

urea group to the naphthyl moiety. Comparing compounds that do not form hydrogen bond with acetate, the spatial distribution of the HOMO in ***m*-2Urea** is mostly localised on the phenyl-group substituents of the naphthalene ring system, and the LUMO orbitals are distributed on the naphthyl rings; the molecular orbitals are not distributed over the urea group (Supplementary Fig. 12). However, in ***p*-2Urea**, the HOMO and LUMO molecular orbitals are distributed around the naphthalene ring system, and it can be seen that the LUMO molecular orbitals are distributed in the urea group. Thus, ***p*-2Urea** can be assumed to have greater overlap between its HOMO and LUMO orbitals than ***m*-2Urea**, having greater oscillator strength on the long-wavelength side of the UV–Vis spectrum (Supplementary Fig. 13 and Supplementary Table 4).

Compared with the molecular orbitals of the hydrogen-bonded acetate complexes, the same tendency is seen for *m*-2Urea, and the molecular orbitals in the hydrogen-bonded complex of *m*-2Urea with acetate are strongly localised to specific substituents (Supplementary Fig. 14). For the HOMO of the hydrogen-bonded complex of *m*-2Urea, no electron density was observed on the naphthalene ring, which suggests that there is weaker absorption in the long-wavelength region for this complex because there is little overlap between the HOMO and LUMO orbitals (Supplementary Fig. 15 and Supplementary Table 5). These calculation results are in agreement with the difference in the shape of the experimental absorption spectra for the two positional isomers on the long-wavelength side.

The emission spectrum of *p*-2Urea in DMSO (150 μM) is characterised by an unstructured broad band centred at 434 nm ($\lambda_{ex}$ = 359 nm), while the emission maxima of *p*-1Urea and *m*-2Urea in DMSO (150 μM) occur at 399 and 423 nm, respectively. These results indicate that the extent of the π-conjugation in *p*-2Urea is greater than those in *p*-1Urea and *m*-2Urea. The addition of TBAAc (up to 250 equiv) induced a substantial quenching of the *p*-2Urea emission[32], while a new emission band developed at lower energies (Fig. 3). It was suggested that the lower-energy band originates from a charge-transfer transition involving a tautomeric form of the FL urea–acetate IHBC in which the N–H proton of the urea groups is transferred from urea to acetate in the excited state[40]. Upon the addition of a large excess of acetate (>250 equiv), a new emission band developed at 448 nm, the intensity of which did not reach a limiting value even at 20,000 equiv. It is worth mentioning that the intensity value of the new emission band at 20,000 equiv is higher than that of the *p*-2Urea emission band before the addition of TBAAc and the emission maxima remain at 448 nm of the fluorescence from IHBC. Therefore, the fluorescence intensity change upon the addition of a large excess of acetate is not caused by the dissociation of IHBC due to high ionic strength which typically affects the binding affinity of host-guest complex[41]. Changes in the emission spectra of *p*-1Urea and *m*-2Urea upon adding excess TBAAc were also investigated, to examine the generality and characteristics of the nonlinear fluorescence intensity change phenomenon. The nonlinear phenomenon was also observed for *p*-1Urea and *m*-2Urea, although the fluorescence intensity increase for *p*-1Urea was minimal, and the value for *m*-2Urea was even smaller compared to that of *p*-2Urea (Fig. 3 and S17). These experimental results indicate that the nonlinear phenomenon upon adding of TBAAc is a general phenomenon for fluorescent urea compounds with a phenylnaphthalene skeleton, and π-conjugation from the hydrogen-bonded urea groups to the naphthyl groups is probably an important factor because of the significant difference between the results for *p*-2Urea and *m*-2Urea.

Thus, we concluded that the nonlinear phenomenon caused by the addition of acetate was a result of the ionic hydrogen bonds between acetate ions and urea groups, which increased the electron-donor character of the urea group and caused intramolecular charge-transfer fluorescence and TICT fluorescence. In general, fluorophores exhibiting TICT fluorescence often display fluorescence intensity increases with increasing solvent viscosity and AIEE[25]. In addition, solutions with an excess of TBAAc are considered to exhibit high viscosity because the acetate anion and tetrabutylammonium cation of TBAAc experience a Coulomb force[42]. Therefore, we investigated the relationship between the increase in fluorescence intensity and the solvent viscosity change when adding an excess of TBAAc.

## Viscosity dependence of fluorescence intensity
First, viscosity increases occurring upon adding TBAAc to DMSO were measured using a viscometer, revealing that the solvent viscosity was increased from 2.3 to 4.7 cP as the amount TBAAc was increased

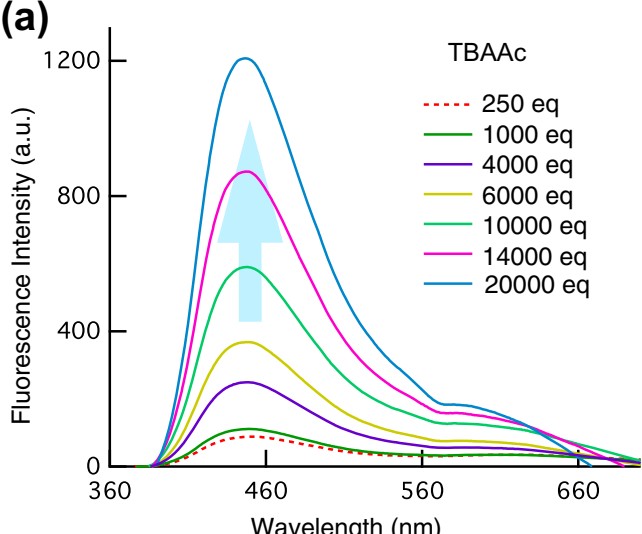

**(a)**

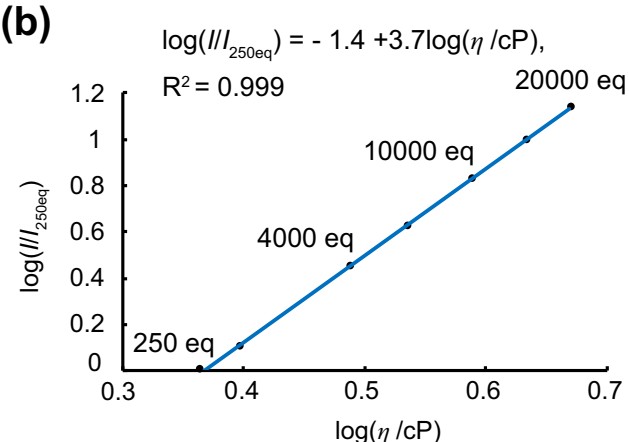

**(b)**

$$\log(I/I_{250eq}) = -1.4 + 3.7\log(\eta/cP),$$
$$R^2 = 0.999$$

**Fig. 4 Changes in fluorescence spectra and fluid viscosity. a** Fluorescence spectra of *p*-2Urea (150 μM, $\lambda_{ex}$ = 359 nm) with various excess amounts of tetrabutylammonium acetate (TBAAc) in DMSO (red dashed = 250 eq; green = 1000 eq; purple = 4000 eq; yellow = 6000 eq; yellowish green = 10,000 eq; pink = 14000 eq; blue = 20,000). **b** Log–log plot of the ratio of fluorescence intensities before and after the addition of ≥250 equiv of TBAAc versus solvent viscosity.

from 250 to 20,000 equiv. The Förster–Hoffmann equation[43,44] describes the relationship between the fluorescence quantum yield and viscosity. To understand this relationship in our system, we plotted the ratio of the FL intensities at 452 nm of *p*-2Urea before and after the addition of ≥250 equiv of TBAAc ($I/I_{250eq}$) versus viscosity (Fig. 4). A linear relationship with a slope of 3.7 ($R^2$ = 0.999) was obtained in the viscosity range from 2.3 to 4.7 cP at 25 °C, as expressed by the following equation (Eq. 1):

$$\log\left(I/I_{250eq}\right) = C + x\log(\eta/cP), \tag{1}$$

where $\eta$ is the viscosity and $C$ is a proportionality constant that depends on concentration and temperature. The slope $x$ is the most important parameter for evaluating the viscosity-sensing capabilities of probe materials because the value is probe-dependent and reflects the degree of change in $I$ with respect to $\eta$. Values of $x$ = 3.7 and $C$ = −1.4 were determined for this molecular system. The torsional motions of the phenyl ring substituents of naphthalene should be inhibited as the viscosity of the media increases. Thus, nonradiative emission decay caused by

this motion the phenyl rings should be effectively suppressed in more viscous media.

Although these effects of the increase in absorbance with an increase in the amount of TBAAc observed in Fig. 2 cannot be ruled out, it should be noted that $x = 3.7$ for the probe-dependent constant is a remarkably high value, compared, for example, with $x = 0.59$ for 9-(dicyanovinyl)julolidine (DCVJ) in a mixture of glycerol and ethylene glycol, a well-known highly viscosity-sensitive fluorescent probe[25,45]. In addition, it should be noted that in this experiment, the ionic strength also changed significantly due to the addition of acetate, and the possibility that the fluorescence intensity changed due to the effect could not be ruled out.

Next, to examine the effects of ionic hydrogen bonding between urea groups and acetate anions, FL measurements of *p*-2Urea in a mixture of glycerol and DMSO, without adding TBAAc, were performed as a control experiment. The relationship between the fluorescence intensity at 434 nm and viscosity was examined with a fixed concentration of *p*-2Urea (1.5 μM) in mixed DMSO–glycerol solvents of various DMSO: glycerol ratios to modulate viscosity [DMSO/glycerol (v/v) = 10/0, 9/1, 8/2, 7/3, 6/4, 5/5, 4/6, 3/7, 2/8, and 1/9] (Supplementary Fig. 18). For the mixed solvents with glycerol amounts up to 80%, the emission intensity at an excitation wavelength of 359 nm increased with viscosity. Indeed, the emission intensity of *p*-2Urea in 80% glycerol was significantly increased, being 1.7-fold greater than that without glycerol. However, in the solvent with a 90% glycerol content, the FL intensity decreased with increasing concentration because of the aggregation of *p*-2Urea, which is insoluble in glycerol. The sensitivity constant of the dye, $x$ in Eq. (1), was found to be 0.12 ($R^2 = 0.971$). Therefore, *p*-2Urea without ionic hydrogen bonding with acetate displayed low viscosity sensitivity. The viscosity sensitivities of IHBCs of *m*-2Urea and *p*-1Urea with acetate ions were also investigated in the presence of excess amounts of TBAAc, and lower viscosity sensitivities than that of *p*-2Urea were obtained (Supplementary Figs. 19 and 20). Further experiments were carried out to investigate temperature sensitivities on the fluorescence intensity of *p*-2Urea with and without TBAAc in DMSO (Supplementary Fig. 21). The logarithm of the fluorescence intensity ratio and the reciprocal of the temperature was plotted, and the slope of the obtained straight line was compared and evaluated as an index of temperature sensitivity (Supplementary Fig. 22). It was found that the temperature sensitivity of the IHBC was higher than that of *p*-2Urea without TBAAc in the temperature range between 25 and 80 °C. In addition, since the maximum fluorescence wavelength of *p*-2Urea on the presence of TBAAc remained about 15 nm longer than on the absence of TBAAc in this temperature range, the luminescent species in the presence of TBAAc is probably the IHBC even at the high temperature.

### Changes in fluorescence properties in solution and the solid-state due to the addition of acetate ions

The absolute fluorescence quantum yields (QYs) in solution, in the absence and presence of TBAAc, were measured to confirm whether *p*-2Urea shows AIEE in the presence of TBAAc. Measurement of FL QYs in the solid-state involves using a powder containing microcrystals obtained by slowly distilling off the solvent and an amorphous state after being ground in a mortar for 10 min, because FL QYs often change depending on the morphology of the FL material, i.e., whether it is in a microcrystalline or amorphous state.

The fluorescence QY of DMSO solution of *p*-2Urea was 18%, while the QYs of powdered *p*-2Urea, which contained microcrystalline and amorphous material, were 7 and 25% before and

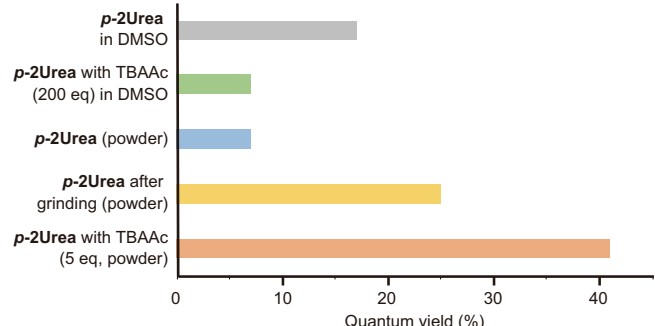

**Fig. 5 Changes in fluorescence quantum yield.** Absolute fluorescence quantum yields of *p*-2Urea powder and *p*-2Urea–DMSO solution in the absence and presence of tetrabutylammonium acetate (TBAAc).

after grinding, respectively (Fig. 5). These unexpected differences between powders before and after grinding may be useful for mechano-stimuli-responsive luminescent materials. The FL QYs of the IHBC of *p*-2Urea and acetate were also measured, in solution and in the solid-state. The FL QY of the IHBC DMSO solution to which 200 equiv of TBAAc had been added, in which the ionic hydrogen bonding between urea groups and acetate ions is considered to be saturated, was revealed to be 7%. However, the solid-state *p*-2Urea–acetate IHBC, prepared by drop-casting the solution of *p*-2Urea with 5 equiv of TBAAc, produced a QY of 41%, which is more than five times higher than that of the solution-phase IHBC. These results indicate that IHBC exhibits AIEE. The IHBC of *p*-2Urea and acetate ions in the solid-state with 5 equiv of TBAAc exhibits an emission red-shift of approximately 18 nm relative to the emission of *p*-2Urea powder in the absence of TBAAc (Fig. 6). In addition, the effects of the amount of TBAAc on the QY of the mixture of *p*-2Urea and TBAAc were investigated by adding 2, 5, 10, 30, 60, 100, and 1800 equiv of TBAAc to the solid-state sample, resulting in FL QYs of 37, 41, 39, 31, 29, 29, and 29%, respectively (Supplementary Fig. 23). These results indicate that the cause of the increase in QY of the IHBC in the solid-state was neither the suppression of intermolecular interactions of the IHBC nor the suppression of self-absorption in the presence of an excess of TBAAc. Furthermore, the effects of the addition of TBAAc to *p*-1Urea and *m*-2Urea were also investigated. In the case of *m*-2Urea, the QY of the DMSO solution was 17% (Supplementary Fig. 24). However, the powder containing microcrystals and the powder after grinding exhibited QYs of 37 and 34%, respectively. Interestingly, *m*-2Urea shows AIEE even without adding TBAAc, which is an opposite result to that of *p*-2Urea. In addition, the DMSO solution of IHBC exhibited 2% QY and the complex with 5 equiv of TBAAc in the solid-state exhibited a QY of 18%. The emission of the *m*-2Urea–acetate ion IHBC in the solid-state is red-shifted by 21 nm relative to the *m*-2Urea powder emission in the absence of TBAAc (Supplementary Fig. 25).

In the case of *p*-1Urea, the DMSO solution without TBAAc shows a high QY of 54%, while the QYs of the powder including microcrystals and the powder after grinding are 44 and 47%, respectively (Supplementary Fig. 26). In addition, the QY of the IHBC of *p*-1Urea and acetate ions was 3% in solution and the QY of the IHBC increased to 50% in the solid-state in the presence of 5 equiv of TBAAc. The *p*-1Urea–acetate ion IHBC in the solid-state exhibits an emission red-shift of approximately 9 nm relative to *p*-1Urea powder in the absence of TBAAc (Supplementary Fig. 27).

These results indicate that IHBC of *p*-2Urea, *p*-1Urea, and *m*-2Urea with acetate ions exhibit much higher QYs in the solid-state than in solution. In particular, the IHBC of *p*-2Urea and *p*-1Urea with acetate ions in the solid-state exhibit higher QYs than those in

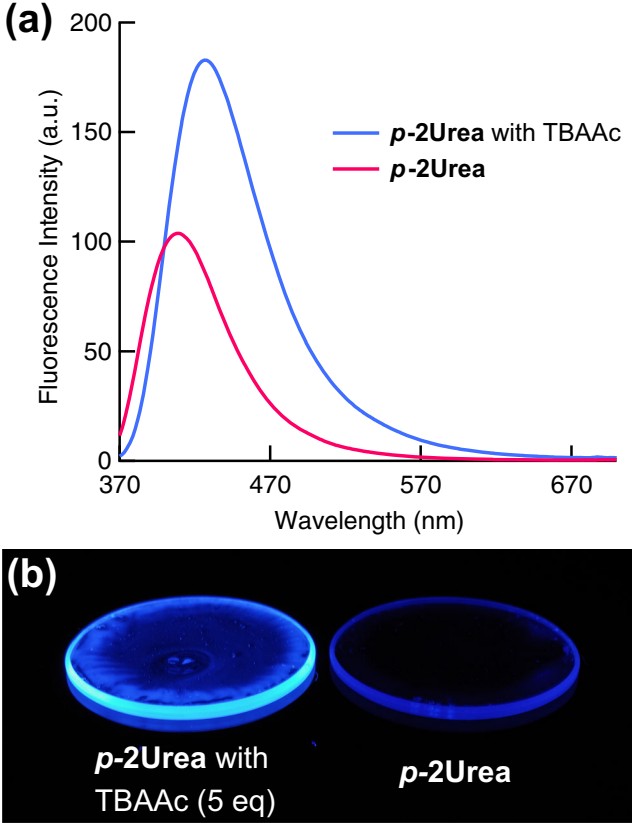

**Fig. 6 Changes in fluorescence with and without tetrabutylammonium acetate (TBAAc). a** Solid-state fluorescence spectra of **p-2Urea** in the presence and absence of 5 equiv of TBAAc when excited at 340 nm. **b** Photographs of the same molar amount of **p-2Urea** with 5 equiv of TBAAc (left) and without TBAAc (right) on 30 mm diameter quartz plates prepared by drop-casting under 365 nm ultraviolet-lamp irradiation.

the absence of TBAAc in the solid-state. In addition, the tendency of the IHBC QY to increase in the solid-state corresponds to enhanced fluid viscosity sensitivity for the fluorescence intensity.

We finally discuss the theoretical origin of the increase in AIEE in the presence of an excess of TBAAc, taking **p-2Urea** as an example. For that purpose, we constructed the simplest models of the aggregate, i.e., two types of dimer models: (i) a naphthalene-naphthalene stacking model, (ii) a urea-urea stacking model (Supplementary Fig. 28). After the DFT-based geometry optimisations including dispersion corrections, we found that type (ii) is more stable than type (i) with or without acetate ions (Supplementary Figs. 29 and 30). For type (ii) in the absence of acetate ions, **p-2Urea**'s are densely packed owing to the π−π and hydrogen-bonding interactions between the whole side chains. On the other hand, in the presence of acetate ions, **p-2Urea**'s are not densely packed due to the electrostatic repulsion but can be stacked by slipping the side chains to avoid the repulsion. The atomic coordinates of the optimised molecular geometries are shown in Supplementary Data 2.

The difference in the aggregation states strongly affects the nature of their excited states. In the absence of acetate ions, the oscillator strength between $S_0$ and $S_1$ is very small: $f \simeq 0.06$ (Supplementary Table 6). In contrast, in the presence of acetate ions, the oscillator strength between $S_0$ and $S_1$ is much larger: $f \simeq 0.21$ (Supplementary Table 7). Such a difference can be explained by molecular orbitals. The electronic transitions between $S_0$ and $S_1$ are ascribed to the HOMO-LUMO and NHOMO-NLUMO transitions in the absence of acetate ions and the HOMO-LUMO transition in the presence of acetate ions, respectively. Both correspond to the charge transfer between urea and naphthalene moieties. In the former case, HOMO and NHOMO are delocalised in the stacking direction, not affecting the π−π conjugation length (Supplementary Fig. 31). In the latter case, HOMO is delocalised over the side chains slipping and contact with each other, leading to the elongation of the π−π conjugation and the enlargement of the transition dipole moment (Supplementary Fig. 32). This is one of the plausible mechanisms for the experimentally observed increase in AIEE in the presence of an excess of TBAAc.

## Conclusions

In summary, we focused on urea groups, which are known to significantly change electron-donating properties by forming strong ionic hydrogen bonds with anion molecules, and investigated the optical properties of neutral fluorescent molecules and IHBCs of fluorescent urea derivatives and acetate ions in both solid-state and solution phases. These results revealed that IHBCs exhibited AIEE and high environmental sensitivities in solution phases. These interesting phenomena show the potential for the control of the characteristics of luminescent dyes by harnessing ionic hydrogen bonding interactions with molecular anions and highlight an approach for the development of solid-state light-emitting materials and environment-sensitive probes. This approach has infinite possibilities since there is an abundance of molecular anions with various structures and functions. A comprehensive investigation on superior urea architectures and molecular anions are currently in progress.

## Methods

**Materials**. All chemicals were purchased from Kanto Kagaku Co., Ltd., TCI chemicals, or Sigma-Aldrich, and used without further purification. Solvents for spectroscopic studies were of spectroscopic grade. Tetrabutylammonium acetate was purchased from Sigma-Aldrich.

**Spectral measurements**. [1]H NMR and [13]C NMR spectra were measured on a Bruker Ascend 500 NMR spectrometer using tetramethylsilane (TMS) as the internal standard. X-ray crystallographic analysis was performed using a Bruker AXS D8 VENTURE/PHOTON 100 diffractometer. UV−Vis spectra were measured using a JASCO V-570 spectrometer. Fluorescence spectra were measured using a JASCO FP-6500 fluorescence spectrometer. Absolute fluorescence quantum yields (Φ) were determined with a JASCO FP-8500 calibrated integrating sphere. ESI-TOF MS data were obtained using a Bruker micrOTOF II mass spectrometer in positive ion mode.

**Typical procedures**

*UV−Vis spectra measurements.* Solutions of **p-2Urea** (DMSO; 0.15 mM) and TBAAc (DMSO; 0.018−1.0 M) were prepared in volumetric flasks under air. The **p-2Urea** solution (0.6 ml) was added to a quartz cell with a 2 mm optical path length, followed by the addition of TBAAc solution (that is, 1 equiv for **p-2Urea**). The UV−Vis spectra were measured before and after the addition of TBAAc. Measurements were repeated after each addition of TBAAc solution.

*Fluorescence spectra measurements in solution.* Solutions of **p-2Urea** (DMSO; 0.15 mM) and TBAAc (DMSO; 0.018−1.0 M) were prepared in volumetric flasks under air. The **p-2Urea** solution (3 ml) was added to a quartz cell with a 10 mm optical path length, followed by the addition of TBAAc solution (that is, 1 equiv for **p-2Urea**). The fluorescence spectra were measured before and after the addition of TBAAc. Measurements were repeated after each addition of TBAAc solution.

*Fluorescence spectra measurements in solid-state.* Solid-state samples were prepared by adding an appropriate amount of a saturated solution of acetate dissolved in chloroform to a fluorescent urea derivative tetrahydrofuran (THF) solution ($1.5 \times 10^{-4}$ M), and then drop-casting the mixture onto a 30 mm diameter quartz plate for vacuum drying.

## Data availability

The X-ray crystallographic coordinates for structures reported in this Article have been deposited at the Cambridge Crystallographic Data Centre (CCDC), under deposition number 2118087. The data generated during this study are included in the published article and the Supplementary Information, the Supplementary Data 1, and the Supplementary Data 2.

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

## Acknowledgements

We would like to thank Mr. S. Ishihara of Instrumental Analysis Centre at Yokohama National University for his excellent technical support for using a fluorescence spectrometer (JASCO FP-8500). We are grateful to Prof. K. Yamamoto, Dr. T. Kambe, Dr. K. Takada (Tokyo Institute of Technology) for their support with measuring ESI-TOF-MS, and Prof. H. Okuzaki at the University of Yamanashi for their support with measuring fluid viscosity. This study was supported, in part, by the Japan Society for the Promotion of Science KAKENHI (Grants-in-Aid for Scientific Research grant no. JP20K15243), Izumi Science and Technology Foundation, and the Cooperative Research Programme of "Network Joint Research Centre for Materials and Devices" for financial support.

## Author contributions

M.T., N.I. and H.N. performed the experiments. N.H. and T.S. conducted the DFT calculations in elucidating the origin of AIEE and analysed the results of the calculations. Y.S. analysed and supervised the single crystal X-ray structure analysis. M.T., N.I., N.H., K.Y., T.S. and M.O. participated in the discussion and writing of the manuscript.

## Competing interests

The authors declare no competing interests.
