## [Peer Review File · Communications Chemistry]

Reviewers' comments:

Reviewer #1 (Remarks to the Author):

In this research work, Takahashi et al. proposed a novel approach to control electronic state of fluorophores and realize AIEE effect using ion hydrogen bonding. Therefore, they synthesized three urea derivatives with D-A structure and built ionic hydrogen-bonded complexes (IHBC) with acetate. By comparing UV absorption and fluorescence spectra, viscosity-sensitive AIEE effect and quantum yields of urea derivatives and their IHBCs in solution and solid state, they concluded that this kind of ion hydrogen bonding played an important role to control of the characteristics of luminescent dyes and realize solid-state emission. Theoretical calculation was also utilized to support experimental phenomenon and above conclusion.

In general, it is a relatively comprehensive study to introduce ionic hydrogen bonds within organic dyes. But there are some problems need to be reconsidered and revised. Hence, it is recommended to accept this paper after careful revision. The suggestions and comments are as follows:

1. The abbreviation of compounds such as "p-2urea" or "p-2Urea" should be consistent throughout the manuscript, supporting information and all figures.
2. In Figure 1c and Supplementary Figure 11&14, these diagrams display two hydrogen bonds between two oxygen atoms of an acetate with two protons of a urea. However, according to theoretical calculation, the optimized structures show only hydrogen bonds between only one oxygen atoms of acetate with two protons of urea, such as p-2urea and m-2urea with two equivalents of AcO⁻. On the contrary, p-1urea with one equivalent of AcO⁻ displays two hydrogen bonds as the same as these diagrams. Therefore, authors should check whether above structural conformations are optimized and then revise above misleading diagrams.
3. In page 2, the author mentioned that "Indeed, tetraphenyl ethane, the most typical AIE dye, having a simple structure, has a low FL QY of 15% in the solid state¹⁷". According to this reference paper, 15% is the QY of aggregates in the water/acetonitrile mixtures with 90% water fraction. And solid sample should be brighter than aggregates. Hence, this description should be checked and revised.
4. In page 2, authors summarized some strategies to realize solid-state emission induced by non-covalent interactions. But they also missed some other methods, such as crystal-state halogen bonding (10.1038/NCHEM.984), intermolecular H Bonding (10.1021/acs.chemmater.8b03699), etc.
5. Frontier molecular orbital of HOMO-1 of p-2urea and m-2urea should be provided to further compare the electronic states and properties of these dyes and their IHBCs, which also gives more information for excitation contributions.
6. Page 8, "The N-H proton peaks were shifted from approximately 8.3 ppm to approximately 11.3 ppm". 11.3 ppm should be about 11.5 or 11.6 ppm, according to Supplementary Figure 10.
7. Page 9, authors pointed out that "The red shift may be attributed to an increase in energy of the HOMO due to the ionic hydrogen bonding between p-2urea and acetate ions." The red shift is usually attributed to smaller energy gap, instead of simple energy change of HOMO.
8. Page 9, "The absorption changes were presumed to be caused by changes in the properties of the solution caused by the addition of excess amounts of TBAAc." What's kind of change in the properties of the solution? It is better to give or proposed a specific change of solution.
9. As discussed in page 19&20, why the QY of powdered p-2urea after grinding is much higher than that before grinding? And why p-1urea shows AIEE effect without adding TBAAc, compared with p-2urea?
10. The QY of p-2urea with TBAAc (20000 eq) could be compared with its pure solution and solid in Figure 5, which may provide information about photophysical property change in different states.

11. Atomic coordinate in Table S7 should be checked and revised, which shows a wrong position of one carbon atom.

12. The format of article and references need to be modified according to the requirements of the journal.

Reviewer #2 (Remarks to the Author):

In the manuscript, the authors investigated a compound containing urea groups that form a strong intermolecular bond with acetate anions. The authors investigate the changes in absorption and fluorescence spectra upon the formation of the ionic complex. They also provide theoretically calculated molecular orbitals and spectra that explain the experimental results. Additionally, they report an increase in the quantum yield of fluorescence when the compound forms a complex with acetate anion in the solid state. I think this is an interesting result that may be important for scientists developing new AIE luminogens, but I am not sure if it will generate wide enough interest to be published in Nat. Commun. Furthermore, I disagree with the authors regarding their claim that their compounds exhibit high sensitivity to viscosity. The fluorescence intensity of p-2urea does increase substantially when TBAAc concentration is increased from 250 eq to 20 000 eq. The latter concentration translates into 3 mol/L, which is a very high concentration. Therefore, going from 250 eq to 20 000 eq the ionic strength of the solution increases massively, yet the authors claim that the fluorescence of p-2urea increases not due to this but due to minute changes in bulk viscosity (2.3 to 4.7 cP). I think this is highly unlikely since viscosity-sensitive probes demonstrate a similar increase in fluorescence intensity when the viscosity is increased by a factor of ~1000. In summary, I recommend rejection.

Reviewer #3 (Remarks to the Author):

The crystal structure is used as proof the authors made the correct compound for their studies, and it certainly does this.

However, the model is not quite complete, CHECKCIF AlertB electron density peak of 1.74 electrons. On investigation there are 2 electron density peak in the different map of around this value, on both the DMSO, that could and should have been modelled as disorder of the Sulphur atom position in the DMSO molecules.

It is standard practice to allow the hydrogen atom involved in the hydrogen bonds to refine freely, in this case they are constrained without an explanation.

Reviewer #4 (Remarks to the Author):

This paper reports the viscosity-sensitive emission of ionic (anionic) hydrogen-bonded complexes (IHBC). The authors succeeded to control the degree of fluorescence with their own synthesized new molecules, also to obtain the results most probably via aggregation induced emission enhancement (AIEE).

Their complexes are new and of course interesting, however, The IHBC as well as the AIEE are not new. For example, the linear relation shown in Fig 4 is of importance, but the physics (equation) is not new.

The discussion on detailed mechanism, why viscosity sensitivity appears, is missing. The most fundamental point of view is the detailed discussion concerning thermodynamics, temperature, entropy, and pressure (density), before the qualitative molecular level arguments such as rotational freedom or DFT orbitals. It is indispensable. Especially, why concentration quenching does not occur? As far as the electronic structure is concerned, the clarification of excited state properties with conical intersection are needed as a function of molecular geometries.

This report must be suitable for more specialized journal.
I do not recommend the publication of this article.

List of Changes

- According to the suggestion by comments 1-12 of reviewer 1, we have unified the abbreviation with ***p*-2Urea**, reflected the hydrogen-bond styles shown in the figure 1c, Supplementary Figures 11, and 14 following the results of computational chemistry, changed the explanation for the reference 17 in page 2, added two new references 20 and 21, added frontier molecular orbitals of HOMO-1, changed the peak value of NMR signal, changed the explanation of the molecular orbital calculation, changed explanations about the part of the increase in absorbance due to the addition of an excessive amount of TBAAc, and added theoretical calculation data about origin of AIEE to our revised manuscript and supplementary information.
- According to the suggestion by comments of reviewer 2, new experiments about temperature dependence in fluorescent intensity were performed and these results were added to Supplementary Figs 21 and 22. In addition, new comments and a reference 41 about both viscosity sensitivity and ionic strength were added to our revised manuscript. Thanks to this experiment, it was found that the fluorescence intensity of our compound greatly changes not only with viscosity but also with temperature. As a result, the viscosity part of the title of our paper was changed to the term “environment” because it is more general and responds to various stimuli.
- According to the suggestion by a comment of reviewer 3, we reanalyzed the X-ray crystallography by treating the sulfur atom in DMSO as a disorder, changed the data of Supplementary Table 1, and submitted the new Checkcif file.
- According to the suggestion by reviewer 4, we have joined two authors who are experts in theoretical chemistry and computational chemistry to unravel the origin of AIEE in our revised manuscript.
- According to the suggestion by comments of reviewer 4, some discussions on the theoretical origin of the increase in AIEE in the presence of an excess of TBAAc and new theoretical calculation data of Supplementary Figs 28-32 and Supplementary Tables 6, 7, 14, 15, 16, and 17 were added to the revised manuscript.

Reply to Reviewer 1

We are very thankful for the quick review and the helpful comments to improve our manuscript.

1. **Comment.** *The suggestions and comments are as follows:*

1. *The abbreviation of compounds such as “p-2urea” or “p-2Urea” should be consistent throughout the manuscript, supporting information and all figures.*

2. *In Figure 1c and Supplementary Figure 11&14, these diagrams display two hydrogen bonds between two oxygen atoms of an acetate with two protons of a urea. However, according to theoretical calculation, the optimized structures show only hydrogen bonds between only one oxygen atoms of acetate with two protons of urea, such as p-2urea and m-2urea with two equivalents of AcO⁻. On the contrary, p-1urea with one equivalent of AcO⁻ displays two hydrogen bonds as the same as these diagrams. Therefore, authors should check whether above structural conformations are optimized and then revise above misleading diagrams.*

3. *In page 2, the author mentioned that “Indeed, tetraphenyl ethane, the most typical AIE dye, having a simple structure, has a low FL QY of 15% in the solid state¹⁷”. According to this reference paper, 15% is the QY of aggregates in the water/acetonitrile mixtures with 90% water fraction. And solid sample should be brighter than aggregates. Hence, this description should be checked and revised.*

4. *In page 2, authors summarized some strategies to realize solid-state emission induced by non-covalent interactions. But they also missed some other methods, such as crystal-state halogen bonding (10.1038/NCHEM.984), intermolecular H Bonding (10.1021/acs.chemmater.8b03699), etc.*

5. *Frontier molecular orbital of HOMO-1 of p-2urea and m-2urea should be provided to further compare the electronic states and properties of these dyes and their IHBCs, which also gives more information for excitation contributions.*

S8, S12, S14

6. *Page 8, “The N–H proton peaks were shifted from approximately 8.3 ppm to approximately 11.3 ppm”. 11.3 ppm should be about 11.5 or 11.6 ppm, according to Supplementary Figure 10.*

7. *Page 9, authors pointed out that “The red shift may be attributed to an increase in energy of the HOMO due to the ionic hydrogen bonding between p-2urea and acetate ions.” The red shift is usually attributed to smaller energy gap, instead of simple energy change of HOMO.*

8. *Page 9, “The absorption changes were presumed to be caused by changes in the properties of the solution caused by the addition of excess amounts of TBAAc.” What’s kind of change in the properties of the solution? It is better to give or proposed a specific change of solution.*

10. *The QY of p-2urea with TBAAc (20000 eq) could be compared with its pure solution and solid in Figure 5, which may provide information about photophysical property change in different states.*

11. *Atomic coordinate in Table S7 should be checked and revised, which shows a wrong position of one carbon atom.*

12. *The format of article and references need to be modified according to the requirements of the*

journal.

Answer. According to the suggestion by comments 1-12 of reviewer 1, we have unified the abbreviation with **p-2Urea**, reflected the hydrogen-bond styles shown in the figure 1c, Supplementary Figures 11, and 14 following the results of computational chemistry, changed the explanation for the reference 17 in page 2, added two new references 20 and 21, added frontier molecular orbitals of HOMO-1, changed the peak value of NMR signal, changed the explanation of the molecular orbital calculation, changed explanations about the part of the increase in absorbance due to the addition of an excessive amount of TBAAC, and added theoretical calculation data about origin of AIEE to our revised manuscript and supplementary information.

2. **Comment.** *As discussed in page 19&20, why the QY of powdered p-2urea after grinding is much higher than that before grinding? And why p-1urea shows AIEE effect without adding TBAAC, compared with p-2urea?.*

Answer. As you pointed out, the changes in fluorescence quantum yield between the crystalline state and the amorphous state of the materials used in this study are very interesting and require further study. However, the results do not affect the purpose or results of this paper, so we plan to study them in the future.

Reply to Reviewer 2

We are very grateful for the quick review and the kind comments about our manuscript.

Comment. *I disagree with the authors regarding their claim that their compounds exhibit high sensitivity to viscosity. The fluorescence intensity of p-2urea does increase substantially when TBAAC concentration is increased from 250 eq to 20 000 eq. The latter concentration translates into 3 mol/L, which is a very high concentration. Therefore, going from 250 eq to 20 000 eq the ionic strength of the solution increases massively, yet the authors claim that the fluorescence of p-2urea increases not due to this but due to minute changes in bulk viscosity (2.3 to 4.7 cP).*

Answer. As you pointed out, ionic strength has a great influence on molecular complex formation [ACS Med Chem Lett. **2014**, 5, 931]. Therefore, you might consider that when an excessive amount of TBAAC is added to the solution the increase in fluorescence intensity is not due to viscosity changes but the dissociation of IHBC due to the increase in ionic strength. However, our experimental facts indicate that the increase in fluorescence intensity due to the addition of excess TBAAC is not due to the dissociation of IHBC due to increased ionic strength. First, before the addition of TBAAC and after the addition of 250 eq, the fluorescence maximum wavelength was extended by 14 nm from 434 nm to 448 nm (Figure 3). From this, the fluorescence wavelength becomes longer when IHBC is formed. Therefore, if the dissociation of IHBC

occurs due to the increase in ionic strength as you pointed out, the maximum wavelength of the newly rising fluorescence should increase while returning to 434 nm. However, in our experimental facts, the rising fluorescence wavelength remains at 448 nm, which is the fluorescence from IHBC. Furthermore, as a second reason, as pointed out in the text, when TBAAC is added at 14000 eq or more, the fluorescence intensity is higher than that before TBAAC is added. If the association is broken and the fluorescence intensity is increased due to the increase in ionic strength, the fluorescence intensity cannot be higher than that before the addition of TBAAC. From these experimental facts, the increase in fluorescence intensity is not due to the dissociation of the IHBC due to the enhancement of ionic strength.

In addition, we conducted an experiment to investigate the temperature dependence of fluorescence intensity before and after adding an excess amount of TBAAC (Supplementary Figures 21 and 22). If the fluorescence intensity changes due to the dissociation of IHBC due to the influence of ionic strength, it is unlikely that the change in fluorescence intensity will change linearly and the sensitivity before and after adding TBAAC to temperature will change. On the other hand, our experimental results showed that the fluorescence intensity changed linearly with temperature, and that *p*-2Urea with an excess amount of TBAAC was more sensitive to temperature than *p*-2Urea without TBAAC. Furthermore, in this experiment, the maximum fluorescence wavelength of *p*-2Urea with an excessive amount of TBAAC remained at 448 nm even when the temperature was raised to 100 °C (Supplementary Figure 21 b). Therefore, the hydrogen bond of IHBC is very strong even with excessive amount of TBAAC and it is difficult to think that increased the fluorescence intensity is due to the dissociation of IHBC due to the increase in ionic strength.

From the experimental facts obtained here, the title of the paper was changed from a specific description of viscosity sensitive to a more general term of environmental sensitive containing temperature dependence.

Figure 3. Difference of Fluorescence maxima between *p*-2Urea and *p*-2Urea-acetate IHBC.

Supplementary Figure 21. (a) Fluorescence spectra of *p*-2Urea (150 μ M, λ_{ex} = 336 nm) in DMSO at different temperatures (298 K to 373 K). (b) Fluorescence spectra of *p*-2Urea (150 μ M, λ_{ex} = 336 nm) with the addition of 14000 equivalents of TBAAc in DMSO at different temperatures (298 K to 373 K).

Supplementary Figure 22. Temperature dependences of fluorescence emissions of *p*-2Urea (150 μ M, λ_{ex} = 336 nm, λ_{em} = 448 nm) with the addition of 14000 equivalents of TBAAc in DMSO and *p*-2Urea (150 μ M, λ_{ex} = 336 nm, λ_{em} = 434 nm) without TBAAc in DMSO at different temperatures (298 K to 373 K).

Reply to Reviewer 3

We are very thankful for the quick review and the helpful comments to improve our manuscript.

Comment. *However, the model is not quite complete, CHECKCIF AlertB electron density peak of 1.74 electrons. On investigation there are 2 electron density peak in the different map of around this value, on both the DMSO, that could and should have been modelled as disorder of the Sulphur atom position in the DMSO molecules.*

Answer. According to your suggestion, we reanalyzed the X-ray crystallography by treating the sulfur atom in DMSO as a disorder, changed the data of Supplementary Table 1, and submitted the new Checkcif file.

Reply to Reviewer 4

Thank you for your constructive suggestion on theoretical aspects.

Comment. *Especially, why concentration quenching does not occur? As far as the electronic structure is concerned, the clarification of excited state properties with conical intersection are needed as a function of molecular geometries..*

Answer. In this revision, we added some discussions on the theoretical origin of the increase in AIEE in the presence of an excess of TBAAc, taking *p*-2Urea as an example. For that purpose, we constructed the simplest models of the aggregate, i.e. two types of dimer models: (i) a naphthalene-naphthalene stacking model, (ii) a urea-urea stacking model (Supplementary Fig. 28).

Supplementary Figure 28. A schematic illustration of dimer models of *p*-2Urea: (a) type (i): a naphthalene-naphthalene stacking model and (b) type (ii): a urea-urea stacking model.

After the DFT-based geometry optimisations including dispersion corrections, we found that type (ii) is more stable than type (i) with or without acetate ions (Supplementary Figs. 29 and 30). For type (ii) in the absence of acetate ions, *p*-2Urea's are densely packed owing to the π - π and hydrogen-bonding interactions between the whole side chains. On the other hand, in the presence of acetate ions, *p*-2Urea's are not densely packed due to the electrostatic repulsion but can be stacked by slipping the side chains to avoid the repulsion.

Supplementary Figure 29. S_0 -optimised structures of the *p*-2Urea dimers in the absence of acetate ions: (a) type (i) and (b) type (ii) with C_i symmetry at the B3LYP/3-21G level of theory with the Grimme's empirical dispersion D3.

Supplementary Figure 30. S_0 -optimised structures of the *p*-2Urea dimers in the presence of acetate ions: (a) type (i) and (b) type (ii) with C_i symmetry at the B3LYP/3-21G level of theory with the Grimme's empirical dispersion D3.

The difference in the aggregation states strongly affects the nature of their excited states. In the absence of acetate ions, the oscillator strength between S_0 and S_1 is very small: $f \approx 0.06$ (Supplementary Table 6). In contrast, in the presence of acetate ions, the oscillator strength between S_0 and S_1 is much larger: $f \approx 0.21$ (Supplementary Table 7). Such a difference can be explained by molecular orbitals. The electronic transitions between S_0 and S_1 are ascribed to the HOMO-LUMO and NHOMO-NLUMO transitions in the absence of acetate ions and the HOMO-LUMO transition in the presence of acetate ions, respectively. Both correspond to the charge transfer between urea and naphthalene moieties. In the former case, HOMO and NHOMO are delocalised in the stacking direction, not affecting the π - π conjugation length (Supplementary Fig. 31). In the latter case, HOMO is delocalised over the side chains slipping and contact with each other, leading to the elongation of the π - π conjugation and the enlargement of the transition dipole moment (Supplementary Fig. 32). This is one of plausible mechanisms for the experimentally observed increase in AIEE in the presence of an excess of TBAAc.

Supplementary Figure 31. Frontier orbitals of the *p*-2Urea dimer (ii) in the absence of acetate ions at the S_0 -optimised structure at the B3LYP/3-21G level of theory with the Grimme's empirical dispersion D3: (a) NHOMO, (b) HOMO, (c) LUMO, and (d) NLUMO. The isosurface value is 1.0×10^{-2} a.u.

Supplementary Figure 32. Frontier orbitals of the *p*-2Urea dimer (ii) in the presence of acetate ions at the S_0 -optimised structure at the B3LYP/3-21G level of theory with the Grimme's empirical dispersion D3: (a) NHOMO, (b) HOMO, (c) LUMO, and (d) NLUMO. The isosurface value is 1.0×10^{-2} a.u.

We finally should add the remark that the above theoretical discussion is based only on the radiative processes of the simple dimer models. As the reviewer pointed out, we further need to clarify the effect of the aggregation not only on the radiative process but also on the nonradiative one through the theoretical search of conical intersections or something like that. However, the nonradiative process is more strongly related to the packing state than the radiative one. We are now planning to carry out large-scale QM/MM calculations based on more realistic aggregate models including many *p*-2Urea's, acetate ions, and their counter ions, and then analyze the radiative and nonradiative processes in detail. It takes an enormous cost and is also out of the scope of this experimental work. We will report it after this work in the near future.

REVIEWERS' COMMENTS:

Reviewer #1 (Remarks to the Author):

The authors have adequately revised their manuscript according to my previous comments and suggestions. The quality of the manuscript has been improved after the revision. I do not have further criticism of the work.

Reviewer #2 (Remarks to the Author):

Ionic strength may easily affect photophysics of IHBC even if dissociation does not take place. For instance, if a charge transfer takes place during non-radiative relaxation, the process is likely to be much faster if ionic strength is increased. I remain with my initial opinion that these results are not enough to prove that IHBC is viscosity-sensitive.

Reviewer #3 (Remarks to the Author):

[Editorial note: This reviewer provided no further comments to the authors.]

List of Changes

- According to editorial requests of the Editorial Requests Table, we have changed a lot of figures, words, setting, typo, and added new sentences to our revised manuscript and there are the details in the right-handed column of the Editorial Requests Table file.
- According to the suggestion by a comment of reviewer 2, a new comment about ionic strength were added to our revised manuscript and we have changed expressions both in abstract and conclusions.

Reply to Reviewer 1

Comment. *The authors have adequately revised their manuscript according to my previous comments and suggestions. The quality of the manuscript has been improved after the revision. I do not have further criticism of the work.*

Answer. We are very thankful for the quick review and the helpful comments to improve our manuscript.

Reply to Reviewer 2

Comment. *Ionic strength may easily affect photophysics of IHBC even if dissociation does not take place. For instance, if a charge transfer takes place during non-radiative relaxation, the process is likely to be much faster if ionic strength is increased. I remain with my initial opinion that these results are not enough to prove that IHBC is viscosity-sensitive.*

Answer. We are very thankful for the quick review and the helpful comments to improve our manuscript. According to your suggestion, we have changed the sentence of abstract from “high sensitivity to fluid viscosity compared with most conventional viscosity-sensitive dyes,” to “**high environmental sensitivities in solution phases**”, changed the sentence of conclusions from “remarkably high sensitivity to fluid viscosity and temperature” to “**high environmental sensitivities in solution phases**”, and added a sentence “**In addition, it should be noted that in this experiment, the ionic strength also changed significantly due to the addition of acetate, and the possibility that the fluorescence intensity changed due to the effect could not be ruled out.**” to our manuscript.